# Immunonutrition in Acute Geriatric Care: Clinical Outcomes, Inflammatory Profiles, and Immune Responses

**DOI:** 10.3390/nu16234211

**Published:** 2024-12-05

**Authors:** Virginia Boccardi, Luigi Cari, Mahdieh Naghavi Alhosseini, Patrizia Bastiani, Michela Scamosci, Giulia Caironi, Giulia Aprea, Francesca Mancinetti, Roberta Cecchetti, Carmelinda Ruggiero, Giuseppe Nocentini, Patrizia Mecocci

**Affiliations:** 1Division of Gerontology and Geriatrics, Department of Medicine and Surgery, University of Perugia, Santa Maria della Misericordia Hospital, 06020 Perugia, Italy; patty_74_it@yahoo.it (P.B.); michela.scamosci@unipg.it (M.S.); giuliacaironi@hotmail.it (G.C.); giuliaprea92@gmail.com (G.A.); francesca_manci@hotmail.it (F.M.); pietroepaolo@hotmail.com (R.C.); carmelinda.ruggiero@unipg.it (C.R.); patrizia.mecocci@unipg.it (P.M.); 2Department of Clinical Pathology, Santa Maria della Misericordia Hospital, 06020 Perugia, Italy; luigi.cari@unipg.it (L.C.); mahdyiehnaghavi@gmail.com (M.N.A.); giuseppe.nocentini@unipg.it (G.N.); 3Division of Clinical Geriatrics, NVS Department, Karolinska Institutet, 17177 Stockholm, Sweden

**Keywords:** aging, frailty, hospital, immunosenescence, nutrition

## Abstract

Background and Aims: Malnutrition is common in acutely ill geriatric patients, worsening immune function and clinical outcomes. Immunonutrition, containing nutrients like omega-3 fatty acids, arginin and glutamine, may improve recovery in this population. This study aimed to evaluate the impact of immunonutrition on clinical outcomes, inflammatory markers, and immune responses in frail, hospitalized older adults. Methods: This is a retrospective observational study. In total, 36 subjects, during hospitalization, received either an immunonutrition formula or isoproteic and isocaloric enteral nutrition. The primary outcome was the length of hospital stay (LOS), with secondary outcomes focused on inflammatory cytokines and immune parameters within a week of hospitalization. Results: Patients were primarily oldest-old, with a mean age of 88.6 years ± 4.9 (range 79–96). The immunonutrition group had a significantly shorter LOS (11.37 ± 4.87 vs. 16.82 ± 10.83 days, *p* = 0.05) and showed increases in key cytokines (G-CSF, INF-α2, IL-12p70, IL-15, IL-2, and IL-3, *p* < 0.05) enhanced immune function. A decrease in T cells and an increased B/T cell ratio was also observed. No significant differences in infection rates or 90-day survival were found. Conclusions: Enteral immunonutrition improved clinical outcomes by reducing LOS and modulating immune responses in frail patients, suggesting potential benefits in recovery. Further studies are needed to confirm these findings.

## 1. Introduction

The aging population is increasingly vulnerable to a range of health challenges, including immune dysfunction, chronic inflammation, and a decline in overall health status. In the context of acute geriatric care, these issues become even more pronounced, often complicating treatment outcomes and recovery processes. In geriatrics, malnutrition continues to be a significant concern, with prevalence rates reaching as high as 50%, varying based on the severity and patient profile, within acute settings [1]. This condition is closely linked to adverse outcomes, such as extended hospitalization, increased disability, and heightened mortality rates [2,3]. Protein-energy malnutrition and subsequent immune system dysfunction frequently afflict older and frail patients suffering from severe medical conditions [4]. The further deterioration in nutritional status correlates with metabolic changes, including a hypermetabolic state and the extent of the inflammatory response induced by acute stress or infection experienced by patients during critical illness [5,6]. Furthermore, acute medical conditions strongly impact a considerable burden on patients, triggering both severe stress and a systemic inflammatory response [7,8].

Immunonutrition, which involves the use of specific nutrients to enhance immune function, has emerged as a promising approach to address these challenges [9]. In recent years, immunonutrition has become a fundamental component in enhanced recovery after surgery and in critically ill patients [10,11,12]. This approach posits that specific nutrients, including amino acids like glutamine or arginine, polyunsaturated fatty acids, such as omega 3, and RNA, whether administered individually or in combination, may serve as immunomodulators [11,13]. Many studies have explored the potential benefits of incorporating omega-3 fatty acids and glutamine into enteral nutrition regimens [14,15], owing to their recognized anti-inflammatory properties [16]. It has been observed that glutamine levels decline during critical illnesses, and these reduced levels are associated with unfavorable clinical outcomes [17]. Notably, enteral supplementation with glutamine has shown promise in enhancing outcomes among burn and trauma patients [18,19].

Enteral nutrition (EN) is advocated for critically ill patients based on recommendations from both American and European intensive care and clinical nutrition organizations [20,21]. EN provides multiple benefits, such as maintaining gut integrity, influencing the systemic immune response, and mitigating the severity of illness. An immune-enhanced EN, which includes specific nutrients aimed at bolstering the body’s response to disease and injury, may also offer added advantages, particularly in frail and geriatric populations; no evidence is available so far in this setting. Frailty is a clinical condition characterized by a decline in physiological reserve and increased vulnerability to stressors, often observed in older adults [22]. Frailty syndrome is a prevalent condition among older adults, particularly those who are hospitalized, and it is closely linked to a range of adverse outcomes. Being frail at the time of hospital admission is itself a significant risk factor for in-hospital mortality, prolonged hospital stays, and a diminished capacity to respond effectively to acute medical events. Moreover, frailty often results in reduced functional abilities upon discharge and is associated with an increased risk of mortality in both the medium and long term [23]. There is currently a lack of evidence supporting the use of enteral immunonutrition in acute geriatric settings, especially when compared to the more substantial body of literature available for other patient populations. While studies have demonstrated the benefits of immunonutrition in various contexts, such as critical care and surgical recovery, its effectiveness in acutely ill older patients remains underexplored and warrants further investigation to establish its clinical value in this population. Considering such evidence, this study aims to investigate the effects of enteral immunonutrition on clinical outcomes, inflammatory markers, and immune responses in an acute geriatric setting of frail hospitalized patients.

## 2. Materials and Methods

### 2.1. Subjects and Study Design

This is a single-center retrospective observational study. Consecutive patients referred to the acute Geriatric Care Unit of Santa Maria della Misericordia Hospital (Perugia, Italy), were assigned, according to good clinical practice, to receive an enteral immunonutrient (IN) or isocaloric isonitrogenous enteral nutrition formula (CTL, control group). Subjects aged 65 years or older able to provide informed consent (or legally authorized representative before enrolment); with acute health issues requiring geriatric care, as determined by the admitting physician; eligible to receive enteral nutrition were included in the study. Patients with known malabsorptive conditions that could affect nutrient absorption; those unable to understand the study’s nature or provide informed consent without a legally authorized representative; patients with terminal organ failure (e.g., liver, kidney, or heart failure) that might confound study outcomes; patients who have recently (<1 month) received immunonutrition supplementation outside the study or subjects undergoing treatment for cancer or with advanced, untreated malignancy, due to potential effects on inflammatory and immune response metrics were excluded. The study was conducted in accordance with the Declaration of Helsinki principles of good clinical practice, and the protocol was approved by the Ethics Committee of the University of Perugia (N: 3979/19)

#### 2.1.1. Primary Objective (Primary Endpoint)

The primary objective of this study was to evaluate the potential effect of immunomodulating enteral nutrition in hospitalized older patients on LOS.

#### 2.1.2. Secondary Objectives (Secondary Endpoint)

The secondary objectives included the study of the effect of enteral immunonutrition on inflammatory and immune system parameters during hospitalization. The mortality at 90 days was also explored.

### 2.2. Study Products

Patients in the IN group were administered an immunomodulating formula (Enteral Impact^®^, Nestle, Osthofen, Germany) daily. This formula is enriched with omega-3 fatty acids, arginine, and dietary nucleotides, components known for their immunomodulatory properties. The control group received the Novasource^®^ GI Control (Nestle, Osthofen, Germany, CTL), supplemented with proteins to ensure an equivalent protein intake between the two groups. This approach was designed to isolate the effects of the immunomodulating components by maintaining consistent protein levels across both experimental and control groups. The metabolic and nutritional needs of participants were assessed based on standard predictive equations commonly used in clinical practice, such as the Harris–Benedict equation or similar tools, adjusted for age, sex, and weight. This approach ensured an estimation of basal energy expenditure, which was further tailored to account for clinical conditions and activity levels, aiming for a normocaloric diet. The decision to utilize a normocaloric diet reflects the goal of maintaining energy balance without overfeeding, which could exacerbate inflammation or metabolic stress in acutely ill older adults. This approach aligns with current guidelines for nutritional support in geriatric populations and hospitalized patients [24]. Below, a comparison of the main nutritional information for the Impact^®^ Enteral and Novasource^®^ GI Control is shown. Impact^®^ Enteral further contains L-arginina 1.3 g, RNA 0.13 g, and Omega 3 0.33 g, per 100 mL. The CTL group received supplementation with Resource Instant Protein to ensure an equivalent daily protein intake as compared to the IN group. Dietary intake data were meticulously documented in the clinic diary, recording the precise daily quantity of nutrients consumed by each patient. This consistent monitoring ensures an accurate assessment of each patient’s caloric and nutritional intake, supporting reliable comparisons between the groups. Table 1 reports the main comparison of nutritional composition between the Impact^®^ Enteral and Novasource^®^ GI Control. Other information can be found on the products website (https://www.nestlemedicalhub.com).

### 2.3. Clinical and Multidimensional Assessment

To avoid the underestimation of a self-rated level of functional capacity, an informant-based rating of functional status was carried out using the Basic Activities of Daily Living (BADL) [25] and the Instrumental Activities of Daily Living (IADL) scales [26]. BADL includes six activities: bathing, dressing, toileting, transferring, continence, and feeding. The IADL includes eight activities: using the telephone, shopping, meal preparation, housekeeping, laundry, use of transportation, self-administration of drugs, and handling finances. Any dysfunction in the performance of these activities was recorded as dependent on the correspondent item. Because IADL items are often gender-specific, we used the version of the scale tested for male subjects that included only five items, with housekeeping, cooking, and laundry excluded. The BADL score ranged from 6 (total independence) to 0 (total dependence), and IADL from 8 (in women) or 5 (in men) (total independence) to 0 (total dependence). The nutritional status was evaluated by administering the Mini Nutritional Assessment (MNA) [27], an instrument developed to detect malnutrition in older subjects. The BRASS (Blaylock Risk Assessment Screening Score) scale is a tool used primarily in the healthcare setting to assess the risk of a patient for early discharge planning [28]. It helps identify patients who may need additional resources or interventions to ensure a safe and effective discharge from the hospital. To calculate the Multidimensional Prognostic Index (MPI) [29], a comprehensive geriatric assessment (CGA) encompassing eight domains was conducted. These domains included the comorbidity index, polypharmacy (number of drugs), presence of pressure ulcers, dependency in basic and instrumental activities of daily living, cognitive function, nutritional status, and social support. The MPI score ranged from 0 (lowest risk) to 1 (highest risk) and was categorized into three risk groups: Low Risk (MPI ≤ 0.33) represents individuals with minimal frailty or no significant geriatric impairments; Moderate Risk (MPI > 0.33 and ≤0.66) indicates a moderate level of frailty or some impairments in geriatric domains. High Risk (MPI > 0.66) reflects significant frailty or multiple severe impairments in geriatric domains associated with a high risk of mortality and poorer health outcomes, often necessitating intensive management and end-of-life care planning.

### 2.4. Analytical Method

Blood samples were collected at the entry into the hospital ward and after a week. Hemoglobin, white blood cells, glucose, total protein, albumin, total cholesterol, transferrin, Vitamin B12, and C-reactive protein (CRP) were determined in serum via routine laboratory methods (Roche Diagnostics, GmbH, Mannheim, Germany). Clearance creatinine was calculated by the BIS-1 (Berlin Initiative Study) formula and expressed as mL/min/1.73 m^2^ [30].

### 2.5. Multiplex Assay

Additional fasting blood samples were collected in EDTA tubes from a peripheral vein in the morning at the entrance of the hospital ward and after a week, and kept immediately on ice at the entry. Plasma was separated via centrifugation (4000 rpm for 15 min at 4 °C), aliquoted, and stored at −80 °C until it was analyzed. A multiplex biometric ELISA-based immunoassay was used according to the manufacturer’s instructions (MILLIPLEX MAP Human Cytokine/Chemokine Magnetic Bead Panel—Immunology Multiplex Assay, Millipore, Burlington, MA, USA). The following molecules were measured and expressed in pg/mL at baseline and after a week of enteral nutrition: EGF, EOTAXIN, G-CSF, GM-CSF, INF-α2, IFN-ɤ, IL-10, IL-12p40 IL-12p70, IL-13, IL-15, IL-17, IL-1RA, IL-1α, IL-1β, IL-2, IL-3, IL-4, IL-5, IL-6, IL-7, IL-8, IP-10, MCP-1, MIP-1α, MIP-1β, TNF-α, TNF-β, VEGF, RANTES. Measurements were performed in duplicate. The analyte concentration was calculated using a standard curve with software provided by the manufacturer (Bio-Plex Manager Software, version 5, Bio-Rad Laboratories, Hercules, CA, USA).

### 2.6. Flow Cytometric Evaluation of Immune System

The staining of CD19 (clone *HIB19*), CD3 (clone *OKT3*), CD4 (clone *OKT4*), and CD8 (clone *OKT8)* surface markers of lymphocytes were performed for 30 min at room temperature, avoiding the light. After washing, 1 mL of PBS with 1% of fetal bovine serum was added, the cells were filtered using pre-separation filters (70 μM—Miltenyi Biotec, Waltham, MA, USA) and the samples were acquired using the Attune NxT flow cytometer (ThermoFisher, Waltham, MA, USA). Data were further analyzed through FlowJo V.10.8.1 (BD Biosciences, Franklin Lakes, NJ, USA). The % of positive cells to the different markers was retrieved. All the monoclonal antibodies used for the staining were purchased from ThermoFisher. In Appendix A, the details of antibodies used for flow cytometry analysis and the gating strategy are shown.

### 2.7. Telomerase Activity in Peripheral Blood Mononuclear Cells (PBMC)

PBMC were isolated by centrifugation in a Lympho-Ficoll (Ficoll-Paque™ PLUS, GE Healthcare) gradient density centrifugation (400× *g* for 30 min). The mononuclear layer was then collected and washed three times in PBS supplemented with 2% FBS (STEMCELL Technologies), and finally, an aliquot containing 2 × 10^5^ cells was transferred into an Eppendorf tube, centrifuged at 3000× *g* for 5 min, and the pelleted cells were preserved at −80 °C until they were used. Telomerase activity in PBMC was measured in duplicate using a telomerase PCR-ELISA (Roche Diagnostics Corp., Indianapolis, IN, USA), based on the telomeric repeat amplification protocol. The intra-assay coefficient of variability was 3.2%, and the inter-assay coefficient of variability was 3.6%. The cutoff for CV for a sample to be repeated was 10%. As a positive control, a cell extract prepared from immortalized telomerase-expressing human kidney cells (293 cells) provided by the kit, was used; for producing negative controls, heat treatment of cell extract for 10 min at 85 °C was performed according to the manufacturer’s recommendations.

### 2.8. Statistical Analysis

The observed data were normally distributed (Shapiro–Wilk test) and are presented as means ± standard deviation (SD). To assess differences among groups, unpaired t-test or Pearson’s Chi-squared (χ2) test were used, as appropriate. The sample size calculation was estimated by GPower 3.1.7 software (http://www.softpedia.com). The total sample size of 36 subjects, estimated post hoc according to a global effect size f^2^ of 0.15 with type I error of 0.05, had a power of 90%. All *p*-values were two-tailed, and the level of significance was set at *p* ≤ 0.05. Statistical analyses were performed using the SPSS 20 software package (SPSS, Inc., Chicago, IL, USA) and PRISM version 9.5.1 (GraphPad Software, Boston, MA, USA).

## 3. Results

### 3.1. Clinical Variables

Between September 2021 and December 2022, over 250 patients were screened, and 36 patients (28 F/8 M), already in enteral nutrition were enrolled in the study and assigned to receive the immunonutrient formula (IN, *n* = 19) or the control isocaloric formula (CTL, *n* = 17). All patients had 100% compliance with the assigned products. No patient received additional nutrients outside the study protocol. The most frequent cause of hospitalization was pneumonia (22.2%), followed by delirium (16.7%) and stroke (16.7%). The least common reason was epileptic seizures (8.3%). Collectively, pneumonia, delirium, and stroke accounted for more than half (55.6%) of the total hospitalizations. No difference in the cause of hospitalization was found between groups (Appendix A). Table 2 reports the total population’s characteristics, which are stratified by the type of nutrition. Patients were primarily oldest-old, with a mean age of 88.6 years ± 4.9 (range 79–96). Most patients were women (with no difference in sex distribution), with severe disability at high risk of malnutrition and in polypharmacotherapy. All subjects were at high-risk MPI (>0.66), reflecting significant frailty or multiple severe impairments in geriatric domains, with no difference between groups. Subjects assigned to immunonutrion had a significant lower length of hospital stay (LOS) as compared to control group. No other difference was found in the other variables explored.

No correlation was found between MPI and LOS (r = 0.402, *p* = 0.110) even after controlling for age and sex (r = 0.444, *p* = 0.098).

Table 3 reports the main biochemical characteristic, and no significant difference was found between groups of nutrition. No difference between groups was found in delirium incidence (47.1% vs. 42.1% χ^2^ = 0.089, *p* = 0.515), infective complications (64.7% vs. 52.6% χ^2^ = 0.538, *p* = 0.347), the appearance of fever (53.3% vs. 46.7% χ^2^ = 0.385, *p* = 0.389), respectively, in control and immunonutrion group. A Kaplan–Meier curve showed a higher survival at 90 days in the IN group, even if such a difference did not result statistically significant (*p* = 0.634).

### 3.2. Plasma Inflammatory Molecules Profile

Among the total population, 23 subjects (18 F/5 M; 13 CTL, and 10 IN) had available inflammatory molecule measurements at two time points during hospitalization. Appendix A report plasmatic levels of inflammatory molecules in the studied population at baseline and after 1 week, stratified by groups of nutrition. Table 4 shows the modulation of plasmatic levels of cytokines in the studied population between the baseline (T0) and after 1 week (T1). Partial correlation analyses, controlled for age and gender, showed no significant correlations between MPI and all inflammatory molecules explored at baseline and after 1 week (*p* > 0.05). A statistically significant increase difference between groups was found in G-CSF, INF-α2, IL-12p70, IL-15, IL-2, and IL-3 levels. An overall increase (Figure 1) was found in the IN groups in G-CSF, INF-α2, IL-12p70, IL-15, IL-2, and IL-3, as compared to control, where instead, a reduction was found.

### 3.3. Immune System Profile

Table 5 and Figure 2 show the change in immune cells during hospitalization. Subjects in the IN group showed a decrease in the T cell percentage and an increase in the B/T cell ratio. Moreover, a slight increase in T cytotoxic cells was observed although not significant (*p* = 0.0543). Detailed immune cells count are reported in the Appendix A.

### 3.4. Telomerase Activity in PBMC

Figure 3 presents the T1/T0 ratio for telomerase activity and illustrates the trend in differences between the two studied groups. Overall, a reduction in telomerase activity is observed in both groups. However, it is important to note that the decrease in telomerase activity following IN administration is not statistically significant, whereas the decrease observed in the CTL group is statistically significant (*p* = 0.006).

## 4. Discussion

This study’s results demonstrate that immunonutrition in an acute geriatric setting can significantly improve short-term clinical outcomes potentially mediated by biological modulations. These improvements include a reduction in the length of hospital stay (LOS) and enhancements in inflammatory and immune markers, highlighting the potential of immunonutrition to positively impact the recovery and overall health of older and frail patients. A recent comprehensive systematic review found that oral or enteral immunonutrition, containing multiple immunonutrients, can reduce morbidity rates in patients undergoing major abdominal surgery. However, its impact on mortality, incidence of infectious diseases, and length of hospital stay remains unclear [31]. Another systematic review in intensive care units showed the same results [32]. No evidence is available in acute geriatric settings [33]. The finding that subjects assigned to immunonutrition had a significantly shorter length of hospital stay (LOS) compared to the control group suggests that immunonutrition may be beneficial in hospitalized patients’ recovery process. This result implies that the targeted use of specific nutrients to enhance immune function could lead to more efficient healing, reduce complications, and potentially expedite discharge. The shorter LOS benefits patients by reducing their exposure to hospital-associated risks, such as infections, and has positive implications for healthcare systems by lowering overall treatment costs and improving bed availability.

Our results indicate that there were no significant differences between the immunonutrition group and the control group in terms of delirium incidence, infective complications, and the appearance of fever. Additionally, although the Kaplan–Meier curve suggested a trend toward higher survival at 90 days in the immunonutrition group, this difference was not statistically significant. These results highlight the complexity of patient responses to nutritional interventions and suggest that the benefits of immunonutrition may not extend uniformly across all clinical outcomes.

From a biological point of view, we found a significant modulation of plasmatic cytokine levels in the studied population, with notable differences observed between the baseline and after one week. Specifically, there was a statistically significant increase in the levels of several cytokines, including G-CSF, INF-α2, IL-12p70, IL-15, IL-2, and IL-3, between the groups. G-CSF is crucial for the proliferation and differentiation of neutrophils, a type of white blood cell important for innate immunity. Elevated G-CSF levels can enhance the body’s ability to fight off infections, suggesting that the intervention or condition being studied effectively mobilizes granulocyte production [34]. IFN-α2 is also part of the innate immune response, particularly in antiviral defense [35]. IL-12p70 is a cytokine that promotes Th1 differentiation and activates NK cells and cytotoxic T lymphocytes. The rise in IL-12p70 could indicate an enhanced capacity to mount a strong cell-mediated immune response, possibly reflecting a response to an infection [36]. IL-15 supports the survival and function of NK cells and memory CD8+ T cells, which are critical for long-term immunity and clearing infected or malignant cells. An increase in IL-15 levels might suggest enhanced immune surveillance and sustained immune memory [37]. IL-2 is key to T cell proliferation and survival and the maintenance of regulatory T cells (Tregs). The observed increase in IL-2 levels might reflect an upregulation of T cell activity, which is crucial for adaptive immunity [38]. Indeed, IL-3 is involved in the growth and differentiation of hematopoietic stem cells into multiple blood cell lineages. Its increase may suggest enhanced hematopoiesis, possibly in response to a need for more immune cells [39]. Moreover, our findings reveal no statistically significant differences in cytokine profiles among the MPI-defined risk groups. This observation suggests that although the MPI is a robust tool for assessing multidimensional frailty and stratifying risk in older adults, it does not appear to directly correlate with variations in inflammatory cytokine levels within this cohort. These results further support the effect of immunonutrition on systemic inflammatory markers.

Our findings suggest that the intervention, likely immunonutrition, had a measurable impact on the immune response, as evidenced by the elevated levels of these cytokines. The cytokines that increased are key players in immune regulation and response, with roles ranging from the stimulation of immune cell production (G-CSF) to the activation and proliferation of T cells (IL-2, IL-15) and the promotion of inflammation (INF-α2, IL-12p70). The significant increases in these cytokines indicate a potential enhancement of the immune system’s ability to respond to stress, which could be particularly beneficial in populations at risk of immunosuppression or in acute settings where robust immune function is critical.

Notably, subjects receiving IN exhibited a decrease in the percentage of T cells, accompanied by an increase in the B/T cell ratio. Additionally, there was a slight, though not statistically significant, increase in T cytotoxic cells. The decrease in T cell percentage could suggest a shift in immune regulation, possibly reflecting a redistribution of T cells or a change in their proliferation rates. This decrease might indicate a temporary suppression or modulation of the T cell-mediated immune response, which could be a response to the specific nutrients provided through immunonutrition. The increase in the B/T cell ratio is particularly interesting as it may indicate a relative increase in B cells or a disproportionate decrease in T cells. This shift could suggest a move toward a more antibody-driven immune response, as B cells are primarily responsible for antibody production. Such a change might be beneficial in contexts where a strong humoral response is required, though it could also reflect a shift away from cellular immunity, which is crucial for combating certain types of infections and cancers. Although not statistically significant, the observed trend toward an increase in T cytotoxic cells suggests a potential enhancement of the immune system’s ability to target and eliminate infected cells. The near significance of this result indicates that with a larger sample size or a longer duration of study, this increase might reach statistical significance, which would have important implications for the potential benefits of immunonutrition in bolstering cellular immunity. Overall, these findings suggest that immunonutrition may influence the balance and composition of immune cells, potentially favoring a more antibody-oriented response while also hinting at an enhanced cytotoxic capacity.

Indeed, telomerase activity, which plays a critical role in cellular aging and the maintenance of chromosomal integrity, was observed to decrease in both groups over time. However, the nature of this decrease differs significantly between the groups and no other study has been conducted so far. In the group receiving IN, the reduction in telomerase activity was not statistically significant, suggesting that IN may help maintain telomerase activity or at least slow its decline. This could imply a potential protective effect of immunonutrition on cellular aging processes, as preserved telomerase activity is often associated with better cellular health and longevity [40]. Conversely, in the CTL group, the decrease in telomerase activity was statistically significant. This significant reduction suggests that, in the absence of immunonutrition, patients may experience more pronounced telomerase reduction, which could be indicative of accelerated cellular aging or a diminished capacity for cellular repair during the stress of hospitalization. The divergence in telomerase activity between the two groups highlights a potential benefit of immunonutrition in mitigating the decline of this critical enzyme. While the exact clinical implications of maintaining telomerase activity are still being explored, it is generally understood that higher telomerase activity is associated with improved cell survival and function [40], which could translate into better overall health outcomes, particularly in vulnerable populations such as hospitalized older patients.

The study also has some limitations. The study’s small sample size reduces statistical power, increases the risk of type II errors, and limits subgroup analyses for variables like acute event type or comorbidities. Its non-randomized, retrospective observational design introduces selection bias and hinders causal inferences between nutrition and outcomes. However, despite the lack of randomization, the groups were homogeneously distributed with no significant differences observed in age, sex, or baseline clinical characteristics. Despite using the Multidimensional Prognostic Index (MPI) to account for frailty, confounders may still obscure the specific impact of nutrition. Variability in individual metabolic needs and the study’s short timeframe may also affect the uniformity and long-term interpretation of results. While the one-week intervention period was sufficient to observe clinical and biological benefits, it is worth considering whether a longer duration of nutritional intervention, potentially extending at home, might yield more pronounced improvements in clinical an immunological outcome. Indeed, our experimental protocol did not include the determination of absolute cell counts immediately following blood sample collection. Instead, peripheral blood mononuclear cells (PBMCs) were isolated using density gradient centrifugation prior to analysis. As a result, the total PBMC count obtained is influenced by the experimental purification process. To characterize immune populations, we employed flow cytometry to analyze the PBMC fraction, reporting percentage values that represent the relative abundance of each cell type. These percentages provide reliable and reproducible data aligned with the objectives of our study. These limitations highlight the need for cautious interpretation and underscore the importance of future studies with larger, randomized cohorts to validate these findings. A comprehensive control of confounders to validate and expand upon the conclusions of this research is also required.

## 5. Conclusions

In conclusion, immunonutrition has effectively modulated clinical outcomes, inflammatory markers, and immune responses in an acute geriatric setting. The observed modulation suggests that tailored nutritional support can play a critical role in enhancing the resilience of older and frail patients by modulating inflammatory responses and strengthening immune function [9]. This approach could be pivotal in improving recovery rates, reducing morbidity, and potentially enhancing the overall quality of life in this vulnerable population. These findings underline the importance of integrating immunonutrition into the standard care protocols for acute geriatric patients to optimize clinical outcomes.

## Figures and Tables

**Figure 1 nutrients-16-04211-f001:**
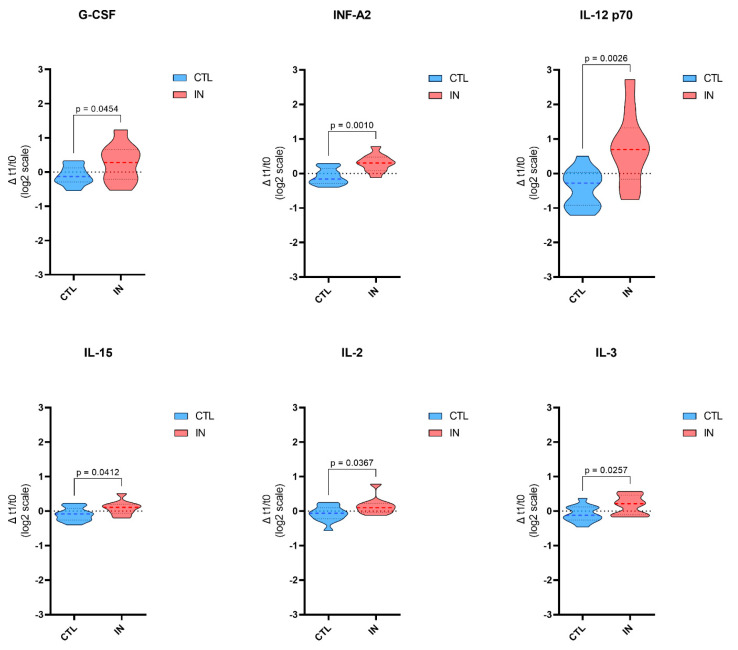
Effect of immunonutrition on cytokine and growth factor levels. This figure shows the changes in cytokine and growth factor levels between the Control (CTL) group and the Immunonutrition (IN) group in a cohort of geriatric patients. The cytokines and factors measured include G-CSF, INF-A2, IL-12 p70, IL-15, IL-2, and IL-3. The data are presented as violin plots, where each plot compares the CTL group (blue) with the IN group (red). Statistically significant differences between the two groups are marked with their respective *p*-values. These results suggest that immunonutrition may play a role in modulating inflammatory and immune responses in this patient population.

**Figure 2 nutrients-16-04211-f002:**
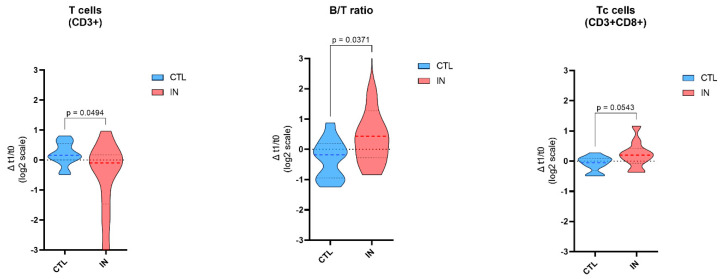
Impact of immunonutrition on T cell bubsets and B/T Ratio. This figure illustrates the changes in T cell (CD3+), B/T cell ratio, and Tc cell (CD3+CD8+) populations between the Control (CTL) group and the Immunonutrition (IN) group in a cohort of geriatric patients. The data are presented as violin plots, with the CTL group shown in blue and the IN group shown in red. Statistically significant differences between groups are annotated with corresponding *p*-values. These findings suggest that immunonutrition may enhance the immune response by modulating T cell populations and altering the B/T ratio in older patients.

**Figure 3 nutrients-16-04211-f003:**
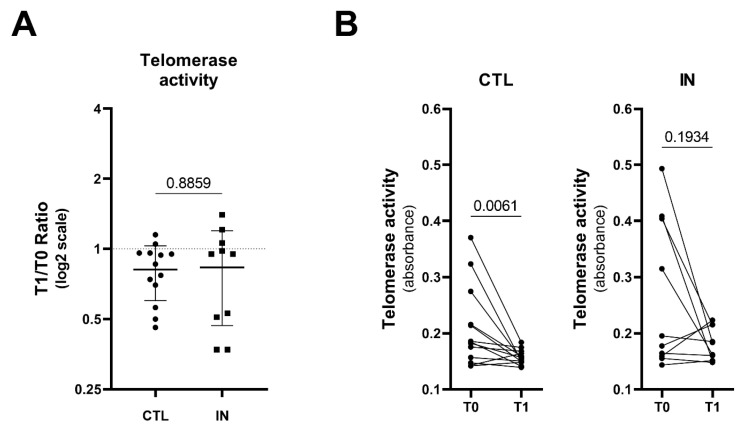
Effect of immunonutrition on telomerase activity. This figure presents the changes in telomerase activity between the Control (CTL) and Immunonutrition (IN) groups. Panel (**A**) shows the T1/T0 ratio of telomerase activity in both CTL and IN groups. There was no significant difference in the T1/T0 ratio between the two groups (*p* = 0.8859), indicating comparable telomerase activity over time. Panel (**B**) depicts individual changes in telomerase activity (absorbance) from baseline (T0) to follow-up (T1) for the CTL and IN groups. In the CTL group, telomerase activity significantly decreased over time (*p* = 0.0061), while the IN group showed a non-significant reduction (*p* = 0.1934), suggesting that immunonutrition may help preserve telomerase activity compared to the control condition.

**Table 1 nutrients-16-04211-t001:** Comparison of main nutritional composition between the Impact^®^ Enteral and Novasource^®^ GI Control (per 100 mL).

Nutrient	Impact^®^ Enteral	Novasource^®^ GIControl
Energy	101 Kcal	110 Kcal
Protein	5.6 g	4.1 g
Carbohydrates	13.4 g	14.0 g
Fat	2.8 g	3.5 g
Saturated Fatty Acids	1.6 g	1.2 g
Monounsaturated Fatty Acids	0.5 g	1.1 g
Polyunsaturated Fatty Acids	0.5 g	0.9 g
Osmolarity	298 mOsm/L	286 mOsm/L
Omega-3 Fatty Acids (EPA/DHA)	0.3 g	-
RNA	0.1 g	-
Arginine	1.3 g	-
Vitamins (selected)		
- Vitamin A	100 µg	125 µg
- Vitamin D	0.67 µg	1.7 µg
- Vitamin E	3 mg	1.7 mg
- Vitamin C	6.7 mg	11 mg

**Table 2 nutrients-16-04211-t002:** Baseline population sample characteristics (*n* = 36).

	Total Sample *n* = 36	CTL *n* = 17	IN *n* = 19	*p*
Sex (F/M)	28/8	14/3	14/5	0.414
Age, y	88.67 ± 4.90	87.47 ± 5.26	89.74 ± 4.42	0.170
ADL	1.14 ± 1.80	1.06 ± 1.91	1.21 ± 1.75	0.813
IADL	0.63 ± 1.91	0.75 ± 2.01	0.53 ± 1.86	0.736
MNA	17.19 ± 4.83	16.25 ± 4.65	17.90 ± 4.98	0.379
BRASS	26.07 ± 3.84	26.43 ± 4.09	25.73 ± 3.71	0.635
MPI	0.72 ± 0.08	0.74 ± 0.11	0.71 ± 0.06	0.536
N drugs	6.06 ± 3.71	6.69 ± 3.89	5.53 ± 3.56	0.364
LOS	13.94 ± 8.57	16.82 ± 10.83	11.37 ± 4.87	0.050

Sex χ^2^ = 0.390. CTL: control; IN: immunonutrition. BADL: Basic Activities of Daily Living; IADL: instrumental activities of daily living; MNA: Mini Nutritional Assessment; BRASS: Blaylock Risk Assessment Screening Score; MPI: Multidimensional Prognostic Index; N drugs: number of drugs; LOS: length of hospital stays.

**Table 3 nutrients-16-04211-t003:** Main biochemical characteristic of the population sample characteristics (*n* = 36).

	CTL *n* = 17	IN *n* = 19	*p*
Hemoglobin (mg/dL)	11.19 ± 2.43	11.56 ± 2.02	0.622
WBC (×10³/µL)	10.75 ± 4.92	12.03 ± 5.70	0.486
Glucose (mg/dL)	121.38 ± 36.41	113.94 ± 38.08	0.577
Total Protein (mg/dL)	5.95 ± 0.58	5.97 ± 0.86	0.937
Albumin (mg/dL)	3.00 ± 0.30	3.09 ± 0.44	0.511
Total-Cholesterol (mg/dL)	155.63 ± 3.93	172.08 ± 67.4	0.527
Transferrin (mg/dL)	207.46 ± 55.25	201.36 ± 60.10	0.786
Vitamin B12 (pg/mL)	325.63 ± 146.76	354.17 ± 288.86	0.572
Folic acid (ng/mL)	10.14 ± 6.49	8.56 ± 5.91	0.500
CRP (mg/L)	7.82 ± 7.25	8.71 ± 8.57	0.751
Clearance creatinine (BIS1)	54.79 ± 27.20	40.03 ± 18.36	0.065

WBC: white blood cells; BIS1: Berlin Initiative Study 1 equation; CRP: C-reactive protein.

**Table 4 nutrients-16-04211-t004:** Modulation of plasmatic levels of inflammatory molecules (expressed in pg/mL) in the studied population between the baseline (T0) and after 1 week (T1).

	CTL *n* = 13	IN *n* = 10	*p*
EGF	0.153 ± 0.464	−0.020 ± 0.511	0.4050
Eotaxin	−0.289 ± 0.495	0.004 ± 0.800	0.2924
G-CSF	−0.105 ± 0.271	0.264 ± 0.547	0.0454
GM-CSF	−0.042 ± 0.266	0.206 ± 0.703	0.3840
INF-α2	−0.085 ± 0.232	0.303 ± 0.253	0.0010
INF-γ	0.107 ± 0.214	0.232 ± 0.631	0.5100
IL-10	0.025 ± 0.415	0.194 ± 0.479	0.3739
IL-12 *p* 40	0.113 ± 0.283	0.035 ± 0.255	0.5015
IL-12 *p* 70	−0.435 ± 0.542	0.710 ± 1.044	0.0026
IL-13	−0.005 ± 0.379	0.033 ± 0.208	0.4186
IL-15	−0.092 ± 0.195	0.089 ± 0.202	0.0412
IL-17	−0.045 ± 0.462	0.358 ± 0.589	0.0799
IL-1 RA	0.020 ± 0.611	0.151 ± 1.688	0.7715
IL-1 α	0.000 ± 0.287	0.127 ± 0.265	0.2921
IL-1 β	−0.092 ± 0.419	0.187 ± 0.723	0.6152
IL-2	−0.074 ± 0.223	0.148 ± 0.253	0.0367
IL-3	−0.085 ± 0.231	0.172 ± 0.284	0.0257
IL−4	−0.001 ± 0.337	0.039 ± 0.227	0.7517
IL-5	−0.084 ± 0.766	0.304 ± 0.660	0.2157
IL-6	−0.3585 ± 0.845	−0.176 ± 1.15	0.6662
IL-7	−0.285 ± 1.280	0.222 ± 0.442	0.4833
IL-8	−0.223 ± 0.534	0.010 ± 0.827	0.4209
IP-10	−0.208 ± 1.170	0.460 ± 1.192	0.1925
MCP-1	−0.288 ± 0.652	0.109 ± 0.935	0.2434
MIP-1 α	−0.003 ± 0.238	0.107 ± 0.245	0.2247
MIP-1 β	−0.102 ± 0.470	0.091 ± 0.669	0.4263
TNF-α	0.006 ± 0.452	0.177 ± 0.739	0.5000
TNF-β	−0.055 ± 0.193	0.085 ± 0.225	0.3350
VEGF	−0.013 ± 0.144	0.125 ± 0.263	0.2017

EGF (Epidermal Growth Factor); G-CSF (Granulocyte Colony-Stimulating Factor); GM-CSF (Granulocyte-Macrophage Colony-Stimulating Factor; INF-α2 (Interferon-α2); INF-γ (Interferon-γ); IL (Interleukin); MCP-1 (Monocyte Chemoattractant Protein-1); MIP-1 α (Macrophage Inflammatory Protein-1 alpha); MIP-1 β (Macrophage Inflammatory Protein-1 beta); TNF-α (Tumor Necrosis Factor-alpha); TNF-β (Tumor Necrosis Factor-beta); VEGF (Vascular Endothelial Growth Factor).

**Table 5 nutrients-16-04211-t005:** Modulation of immune system cell percentage in the blood of studied population, between the baseline (T0) and after 1 week (T1).

	CTL *n* = 13	IN *n* = 10	*p*
B cells (CD19+)	−0.118 ± 0.507	0.112 ± 1.024	0.4871
T cells (CD3+)	0.190 ± 0.386	−0.553 ± 1.215	0.0494
B-T cell ratio	−0.305 ± 0.653	0.668 ± 1.397	0.0371
T helper cells (CD3+CD4+)	0.004 ± 0.336	−0.072 ± 0.493	0.6649
T cytotoxic cells (CD3+CD8+)	−0.084 ± 0.248	0.216 ± 0.451	0.0543
CD4-CD8 T cell ratio	0.088 ± 0.440	−0.288 ± 0.932	0.2126

## Data Availability

The datasets used and/or analyzed during the current study will be available from the corresponding author on reasonable request.

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
