# Peer review of "Immunonutrition in Acute Geriatric Care: Clinical Outcomes, Inflammatory Profiles, and Immune Responses"

_nutrients, 2024, doi:10.3390/nu16234211_

Round 1
Reviewer 1 Report
Comments and Suggestions for Authors
Thank you for the opportunity to review this manuscript, which addresses the impact of immunonutrition on clinical outcomes and immune responses in hospitalized frail older adults. The study explores an important topic, as malnutrition in acutely ill geriatric patients is a significant factor impacting immune function and clinical recovery.
Given the observational nature and small sample size(only 36 participants.), it would be helpful to more explicitly address potential limitations, including the lack of randomization and the retrospective design. Including a limitations section that discusses these factors, along with potential confounders (e.g., baseline nutritional status or comorbidities), could further strengthen the study's validity.
Author Response
We express our sincere gratitude to the Reviewer for the valuable suggestions and feedback, which have greatly contributed to the revision and improvement of our manuscript.
We appreciate your thorough review and insightful feedback on our manuscript. Your recognition of the significance of malnutrition in acutely ill geriatric patients and its impact on immune function and clinical recovery highlights the relevance of our study. We acknowledge the importance of addressing the limitations of our work, and we have made the following revisions based on your suggestions:
- Addressing limitations:
In response to your comment regarding the observational nature and small sample size (36 participants), we have added a dedicated limitations section in the discussion. - Integration of the MPI:
While the study already considers the Mini Nutritional Assessment (MNA) to evaluate the risk of malnutrition, we have also incorporated the Multidimensional Prognostic Index (MPI) to provide a more comprehensive assessment of participants' baseline conditions. The MPI complements the MNA by integrating additional domains such as comorbidity, functional and cognitive status, and social factors, offering a multidimensional perspective on frailty and vulnerability in our study population. No difference was found between group and reported in the revised manuscript. - Strengthening Study Validity:
We have elaborated on potential confounders, including variations in nutritional and functional status at baseline, and their possible influence on the observed outcomes.
We hope these revisions address your concerns and enhance the overall rigor and transparency of our manuscript. Thank you again for your constructive feedback, which has significantly contributed to the improvement of our work.
Reviewer 2 Report
Comments and Suggestions for Authors
1. Please include relevant references from last 5 years.
2. Please provide aditional analysis about clinical medical conditions of patients, with potential impact on inflamatory profile, evolution trends, lenght of stay.
3. Please compare similar groups of patients (homogenous groups regarding the acut event type, hemodynamic stability, signs of shock, diabetis, recent surgery, trauma) from the point of view of nutrition
4. Please comment how this factors could modify the inflamatory profile and how relevant is only nutrition interpretation as a variabile factor
5. Please discuss how did you calculate/appreciate the methabolic nutritional needs( because you are discussing a normocaloric diet
6. Please comment the potential error factors and their impact on the study results
Author Response
We express our sincere gratitude to the Reviewer for the valuable suggestions and feedback, which have greatly contributed to the revision and improvement of our manuscript.
We appreciate your thorough review and insightful feedback on our manuscript
- As suggested, more recent references have been carefully incorporated into the manuscript. See the revised version.
- We appreciate your valuable suggestion to include additional analyses on the clinical medical conditions of patients and their potential impact on results. In response, we have incorporated this analysis into the revised manuscript, including MPI an supplemental data. These updates are clearly presented in the revised version for your review. Thank you for this constructive feedback, which has strengthened the depth and clinical relevance of our study.
- Thank you for your insightful suggestion to compare similar groups of patients based on acute event type, hemodynamic stability, presence of shock, diabetes, recent surgery, or trauma in relation to their nutritional status. Due to the small sample size (36 participants), conducting such stratified subgroup analyses was not statistically feasible without compromising the robustness of the findings. Instead, to address the concern regarding the uniformity of data, we have included the Multidimensional Prognostic Index (MPI) and the primary reasons for hospitalization as key parameters. These metrics provide a comprehensive overview of the patients’ baseline characteristics and acute conditions, demonstrating the relative homogeneity of the cohort in terms of frailty and clinical status.This approach allows us to present a balanced perspective on the nutritional assessment and its association with clinical outcomes, while acknowledging the inherent limitations posed by the sample size. We have detailed this explanation in the revised manuscript to clarify the methodology and rationale for our approach. Thank you for your understanding and for the opportunity to refine our study.
- In the revised manuscript, we have addressed the analysis of the correlation between the Multidimensional Prognostic Index (MPI) and cytokine levels. Our findings indicate that there were no statistically significant differences in cytokine profiles across MPI risk groups. This suggests that while the MPI effectively captures the multidimensional frailty and risk status of older adults, it may not directly correlate with variations in inflammatory cytokines in this specific cohort. We have included this observation in the results and discussion sections, emphasizing the need for further studies with larger sample sizes to explore potential subtle or indirect associations between frailty indices like the MPI and immune-inflammatory markers. This addition highlights the complexity of the interplay between frailty and systemic inflammation. By the way these results further support the effect of immunonutrition on systemic inflammatory markers.
- In our study, the metabolic and nutritional needs of participants were assessed based on standard predictive equations commonly used in clinical practice, such as the Harris-Benedict equation or similar tools, adjusted for age, sex, and weight. This approach ensured an estimation of basal energy expenditure, which was further tailored to account for clinical conditions and activity levels, aiming for a normocaloric diet. The decision to utilize a normocaloric diet reflects the goal of maintaining energy balance without overfeeding, which could exacerbate inflammation or metabolic stress in acutely ill older adults. This approach aligns with current guidelines for nutritional support in geriatric populations and hospitalized patients. Further details have been included in the revised manuscript for clarity.
- Potential error factors in this study stem from both methodological and clinical variables, which could influence the reliability and generalizability of the results. Below is a discussion of these factors and their potential impact: 1) The limited number of participants (36) reduces statistical power and increases the risk of type II errors, potentially masking significant associations. It also limits the ability to perform subgroup analyses, which could better delineate the impact of specific variables like acute event type or comorbidities.2) The non-randomized, retrospective nature of the study introduces selection bias and limits the ability to establish causal relationships between nutritional interventions and clinical outcomes.3) While tools like the Multidimensional Prognostic Index (MPI) were used to account for frailty, these confounders could still obscure the specific impact of nutrition. 4) Individual variations in metabolic and nutritional requirements, influenced by age, activity levels, and acute illness severity, could lead to inaccuracies in meeting patients' needs, affecting the uniformity of the intervention. 5) The study's timeframe may not fully capture long-term trends in nutritional impact or recovery, limiting the understanding of sustained benefits or challenges. These potential errors may have led to an underestimation or overestimation of the effects of immunonutrition on clinical outcomes and inflammatory responses. They underscore the need for caution in interpreting the findings and highlight the importance of conducting future studies with larger, randomized cohorts and comprehensive control of confounders to validate and expand upon the conclusions of this research. These points have been addressed in the revised manuscript.
Reviewer 3 Report
Comments and Suggestions for Authors
1. What were the reasons for hospitalization? Did they differ between the groups?
2. Authors should provide a direct comparison of the nutritional treatment that patients receiverd - this should include values for novasource+ supplement (in other words, this should provide data regarding the final formula that was administered). Moreover, authors should provide more information than is currently present in Table 1 e.g. vitamin, omega acids etc. Finally, a direct information on the dosage should be provided.
3. Flow cytometry: name, clone, fluorochrome, manufacturer and catalogue number have to be clearly listed in a separate table. Moreover, a gating strategy should be provided as a separate figure. Please see the miflowcyt guidelines for reference.
4. How were the patients assigned to the IN/CTL group?
5. I would suggest calculating absolute values for T, Th, Tc, B cells.
6. How many patients were included in flow cytometry and telomerase experiments?
7. Figure title and caption should not be separated
8. Lines 336 - 355 are mostly speculatory and lack direct support in data. As already mentioned, authors should calculate absolute values. Nevertheless, authors should withstand the temptation of such speculations. Instead, please compare it to other similar nutritional studies.
Author Response
We express our sincere gratitude to the Reviewer for the valuable suggestions and feedback, which have greatly contributed to the revision and improvement of our manuscript.
- Thank you so much for this observation. A supplemental table (S2) has been added to the revised manuscript, detailing the reasons for hospitalization across the groups. The analysis showed no significant differences in hospitalization reasons between the groups, indicating comparable clinical conditions at baseline.
- We appreciate the reviewer’s valuable suggestion to include additional information regarding the nutritional products. In response, we have incorporated further details about the products into the revised manuscript. Additional product-specific information can also be accessed on the respective manufacturers' websites, as referenced in the manuscript. The metabolic and nutritional requirements of participants were evaluated using standard predictive equations widely utilized in clinical practice, such as the Harris-Benedict equation, with adjustments made for age, sex, and weight. This method provided an estimation of basal energy expenditure, which was subsequently modified to reflect individual clinical conditions and activity levels, ensuring the provision of a normocaloric diet tailored to each participant's needs. All these details have been thoroughly addressed and are now included in the revised manuscript for clarity and completeness. Thank you again for this constructive feedback, which has helped us improve the quality of our work.
- Thank you for your feedback. In response to your suggestions, we have made the following updates: Supplemental Table 1: We have included a comprehensive table listing the details of the antibodies used for flow cytometry, including the name, clone, fluorochrome, manufacturer, and catalog number. Figure S1: A detailed gating strategy has been provided as a supplemental figure. These additions have been included in the supplementary materials of the revised manuscript. We appreciate your valuable input, which has helped us enhance the clarity and rigor of our work.
- Thank you for your comment. Patients were not randomized due to the observational nature of the study. However, the assignment to the intervention (IN) or control (CTL) groups was conducted following Good Clinical Practice (GCP) guidelines. Despite the lack of randomization, the groups were homogeneously distributed with no significant differences observed in age, sex, or baseline clinical characteristics. This ensured a balanced comparison and maintained the integrity of the observational study design. These details have been clarified in the revised manuscript.
- Unfortunately, our experimental protocol does not include the determination of absolute cell counts immediately after blood sample collection. Specifically, we isolate PBMCs using density gradient centrifugation before proceeding with any analyses, and the total PBMC count we obtain is inherently influenced by the experimental purification procedure. As a result, absolute values cannot be reliably determined in our experimental flow. Instead, we rely on flow cytometry to analyze the immune populations within the PBMC fraction, providing percentage values that reflect the relative abundance of each cell type. These percentages offer robust and reproducible data for our study objectives. We have now included this information in the revised manuscript as a supplemental table.
- As already reported, 23 patients were included in the flow cytometry and telomerase experiments, with available data collected at two time points.
- As kindly suggested, the figure and caption have been combined.
- We acknowledge the reviewer’s insightful comments. As noted, we do not have absolute values or sufficient prior studies that directly enable a comparison with our data. We agree that this represents a limitation of our study, which we have explicitly acknowledged in the revised manuscript. To address the concern regarding speculation in lines 336–355, we have revised this section to ensure that our interpretations remain grounded in the data presented. Indeed, detailed immune cells count are reported in the supplementary materials (see attached excel file). These revisions aim to enhance the rigor and objectivity of the discussion section while acknowledging the study's limitations transparently. Thank you for guiding us to improve the scientific quality of our work.
Round 2
Reviewer 2 Report
Comments and Suggestions for Authors
Thank You for extensive responses and improving the quality and clarity of the study.